# Varying Cellular Immune Response against SARS-CoV-2 after the Booster Vaccination: A Cohort Study from Fukushima Vaccination Community Survey, Japan

**DOI:** 10.3390/vaccines11050920

**Published:** 2023-04-29

**Authors:** Yuta Tani, Morihito Takita, Yurie Kobashi, Masatoshi Wakui, Tianchen Zhao, Chika Yamamoto, Hiroaki Saito, Moe Kawashima, Sota Sugiura, Yoshitaka Nishikawa, Fumiya Omata, Yuzo Shimazu, Takeshi Kawamura, Akira Sugiyama, Aya Nakayama, Yudai Kaneko, Tetsuhiko Kodama, Masahiro Kami, Masaharu Tsubokura

**Affiliations:** 1Medical Governance Research Institute, Tokyo 108-0074, Japan; tyuta0430@gmail.com (Y.T.);; 2Department of Radiation Health Management, Fukushima Medical University, Fukushima 960-1295, Japan; 3Department of General Internal Medicine, Hirata Central Hospital, Fukushima 963-8202, Japan; 4Department of Laboratory Medicine, Keio University School of Medicine, Tokyo 160-0016, Japan; 5Department of Internal Medicine, Soma Central Hospital, Fukushima 976-0016, Japan; 6Proteomics Laboratory, Isotope Science Center, The University of Tokyo, Tokyo 113-0032, Japan; 7Laboratory for Systems Biology and Medicine, Research Center for Advanced Science and Technology, The University of Tokyo, Tokyo 153-8904, Japan; 8Medical and Biological Laboratories Co., Ltd., Tokyo 105-0012, Japan

**Keywords:** SARS-CoV-2, vaccination, booster, cellular immunity, population level

## Abstract

Booster vaccination reduces the incidence of severe cases and mortality related to COVID-19, with cellular immunity playing an important role. However, little is known about the proportion of the population that has achieved cellular immunity after booster vaccination. Thus, we conducted a Fukushima cohort database and assessed humoral and cellular immunity in 2526 residents and healthcare workers in Fukushima Prefecture in Japan through continuous blood collection every 3 months from September 2021. We identified the proportion of people with induced cellular immunity after booster vaccination using the T-SPOT.COVID test, and analyzed their background characteristics. Among 1089 participants, 64.3% (700/1089) had reactive cellular immunity after booster vaccination. Multivariable analysis revealed the following independent predictors of reactive cellular immunity: age < 40 years (adjusted odds ratio: 1.81; 95% confidence interval: 1.19–2.75; *p*-value: 0.005) and adverse reactions after vaccination (1.92, 1.19–3.09, 0.007). Notably, despite IgG(S) and neutralizing antibody titers of ≥500 AU/mL, 33.9% (349/1031) and 33.5% (341/1017) of participants, respectively, did not have reactive cellular immunity. In summary, this is the first study to evaluate cellular immunity at the population level after booster vaccination using the T-SPOT.COVID test, albeit with several limitations. Future studies will need to evaluate previously infected subjects and their T-cell subsets.

## 1. Introduction

Vaccination has been very effective in preventing severe cases of infection from severe acute respiratory syndrome coronavirus 2 (SARS-CoV-2), greatly benefiting the worldwide population [1]. Booster vaccination reduced the incidence of breakthrough, severe cases, and mortality related to SARS-CoV-2 [2,3,4,5,6,7]. Providing booster vaccination to health care workers (HCWs) and people at high risk of experiencing severe cases of infection is currently a global consensus [8].

Cellular immunity plays an important role in protecting individuals against severe cases of COVID-19, as reported in various studies, animal experiments [9,10], clinical studies [11,12,13,14,15,16,17,18,19,20], and general reviews [21,22,23]. Several studies showed that booster vaccination induced stronger cellular immunity among a relatively small number of participants or HCWs [24,25,26,27]. However, to this day, few studies have been conducted at the population level regarding cellular immunity after booster vaccination, and little is known about what proportion of population has achieved induced cellular immunity after booster vaccination and who is more likely to achieve it.

Since September 2021, we have assessed humoral and cellular immunity in 2526 residents and HCWs of Soma, Minami-Soma, and Hirata village in Fukushima Prefecture in Japan via continuous blood collection every 3 months. In addition to the blood analysis data, we have obtained detailed profiling data, including medical history and medication, as well as adverse reactions after vaccination, and pre-infection data on all participating individuals. Therefore, the Fukushima cohort served as a unique database (Fukushima cohort database) [28,29,30,31,32,33] that enabled us to evaluate cellular immunity at the population level after booster vaccination.

This study identified the proportion of people with induced cellular immunity against SARS-CoV-2 in the Fukushima cohort database and analyzed their background characteristics. To evaluate SARS-CoV-2-specific T-cell immune memory, there are methodologies to detect cytokine (especially IFN-gamma) production in antigen-stimulated short-term cultures in vitro, such as the ELISpot assay, QuantiFERON assay [34], and analysis of intracellular expression via flow cytometry, as well as to detect activation marker expression, such as flow cytometry. Herein, we used the T-SPOT.COVID test, which is a standardized ELISpot interferon–gamma release assay. We also investigated the cellular immunity of previously infected individuals after booster vaccination.

## 2. Methods

### 2.1. Ethics Statement

The study was approved by the ethics committees of Hirata Central Hospital (number 2021-0611-1) and Fukushima Medical University School of Medicine (number 2021-116). Written informed consent was obtained individually from all participants before the survey. The experiments were performed in accordance with relevant guidelines and regulations, including the Declaration of Helsinki.

### 2.2. Vaccination Schedules, Participant Eligibility, and Sample Collection

Participants were recruited from Hirata village in Fukushima Prefecture in Japan as a part of the Fukushima cohort study [28,29,30,31,32,33]. In Hirata village, individuals were administered the first and second doses of the BNT162b2 vaccine (Pfizer/BioNTech, New York, NY, USA) during two periods: 6 April to 12 August 2021 for the initial dose, and 27 April to 8 September 2021 for the subsequent dose. The third dose with either BNT162b2 (Pfizer-BioNTech) or mRNA1273 (Moderna, Cambridge, UK) was administered between 23 December 2021 and 3 April 2022.

Participants eligible for this study were those who completed a booster vaccination and had a valid T-SPOT.COVID test result.

Blood samples were collected once during each designated period: June, September, and December 2021, as well as in March 2022. Blood collection (8 mL) was performed at medical facilities. The whole blood and serum samples were sent to the University of Tokyo (Tokyo, Japan) to measure SARS-CoV-2 specific antibodies and cellular immunity.

Information on age, sex, blood type, days between vaccination and blood collection, vaccine type, smoking, drinking habits, commodities, medication, and adverse reactions after vaccination was retrieved from the paper-based interview sheet (summarized in Appendix A).

### 2.3. Measurement of SARS-CoV-2-Specific Antibodies

All serological assays were conducted at the University of Tokyo. Specific IgG (i.e., IgG(S)) and neutralizing activity were measured as indicators of humoral immune status after vaccination against SARS-CoV-2. Chemiluminescent immunoassays using iFlash 3000 (YHLO Biotech, Shenzhen, China) and iFlash-2019-nCoV series (YHLO Biotech) reagents were used in the present study. The lower limit of neutralizing antibody titer (≥500 AU/mL) was defined as 500 AU/mL. All testing processes followed the official guidelines. Quality checks were conducted every day before measurements were recorded.

### 2.4. Measurement of Cellular Immunity

Blood samples for the T-SPOT.COVID test (Oxford immunotec, Abingdon, Cambridge, UK), which is a standardized ELISpot interferon–gamma release assay, were processed and analyzed according to the manufacturer’s instructions. Briefly, blood samples were drawn into lithium heparin tubes that were shipped overnight to Oxford Immunotec in temperature-controlled shipping boxes. Next, the T-Cell Xtend reagent (Oxford Immunotec) was added to samples, peripheral blood mononuclear cells (PBMCs) were isolated through density gradient centrifugation, washed, and counted, and 250,000 cells/well were plated into 4 wells of a 96-well plate. In all cases, 10 mL of peripheral blood was obtained from each individual in lithium heparin tubes, along with an identical volume sample in EDTA tubes from the same individuals.

### 2.5. Statistics Analysis

A descriptive analysis was conducted. Categorical variables were summarized as frequencies and percentages, whereas antibody titers were summarized as medians and interquartile ranges. A multivariable linear regression analysis was performed to determine factors associated with age, sex, blood type, days between booster vaccination and blood collection, vaccine type, smoking, drinking habits, comorbidities, and adverse reactions after the second and third vaccination dose. A *p*-value of ≤0.05 was considered statistically significant. The IBM SPSS Statistics (IBM ver. 28.0.1.0, New York, NY, USA) software was used for all analyses.

## 3. Results

### 3.1. Population Characteristics

A total of 1089 individuals participated in this study. Five (0.5%) had a history of COVID-19 infection. In detail, 1432 individuals participated in the second blood sampling between 16 September and 7 October 2021, 1380 individuals participated in the third blood sampling between 25 November and 25 December 2021, and 1312 and 1121 individuals participated in the fourth blood sampling and in T-SPOT.COVID testing between 8 March and 31 March 2022. The median (interquartile range) interval between the first and second vaccination dose was 21 d (21–21), whereas that between the second and third vaccination dose was 231 d (218–241). The median age of participants and number of women were 56 (35–61) and 737 (67.5%), respectively. Notably, 682 (62.5%) participants received homologous vaccination (BNT162b2) (Table 1). The detail characteristics and information on participants stratified via reactive interpretation of T-SPOT.COVID testing are provided in Appendix A.

### 3.2. Distribution of T-SPOT.COVID Test

We observed that 64.3% (700/1089) of participants had reactive results in the T-SPOT.COVID test, whereas 22.9% (250/1089) had non-reactive results and 12.8% (139/1089) had borderline results. We detected that the median number of spots was 11 [5,6,7,8,9,10,11,12,13,14,15,16,17,18,19,20,21,22,23,24,25]; the distribution of spot numbers is shown in Panel A (Figure 1). Details of the frequency of non-reactive, reactive, and borderline results are shown in Appendix A. We found that the percentages of individuals with non-reactive results (non-reactive group) were 37.2% vs. 55.2% (40–65 years old vs. over 65 years old), 39.6% vs. 60.4% (men vs. women), 30.8% vs. 36.0% (within 30 d vs. over 60 d; days from booster vaccination), 52.4% vs. 47.6% (BNT162b2 vs. mRNA-1273), 72.8% vs. 27.2% (at least one vs. none; comorbidities), 24.8% vs. 20.6% (at least one vs. none; medication), and 49.3% vs. 36.2% vs. 14.8% (none vs. once vs. both; adverse reactions after second and third vaccination dose) (Table 1). Appendix A shows the detailed characteristics of patients in the non-reactive group; 18.5% experienced hypertension, 21.6% experienced heart diseases, and 21.3% experienced diabetes mellitus.

### 3.3. Distribution of IgG(S) and Neutralizing Antibody Titer

We found that the median anti-SARS-CoV-2-IgG (IgG(S) antibody titer) and neutralizing antibody titers were 2212 (1373–3962) and 500 (500–500), respectively (Figure 1B,C). In particular, we detected that 94.7% (1031/1089) and 93.4% (1017/1089) of participants had IgG(S) and neutralizing antibody titers of ≥500 AU/mL. Notably, despite having IgG(S) and neutralizing antibody titers of ≥500 AU/mL, 33.9% (349/1031) and 33.5% (341/1017) of participants, respectively, did not have reactive results in the T-SPOT.COVID testing. We found that 72.8% (182/250) of individuals in the non-reactive group and 80.6% (25/31) with IgG antibody titer of ≤500 (AU/mL) had at least one comorbidity (Appendix A). We detected significant differences in the IgG(S) antibody titer between individuals in the non-reactive and borderline groups and those in the reactive group; the medians were 1687 (1020–2889) and 2674 (1608–4418), respectively (*p* < 0.001, Appendix A).

### 3.4. Logistic Regression Analysis of T-SPOT.COVID Test

Using a univariate analysis, we found that participants aged ≤40 years [OR: 1.86, 95% CI: 1.25–2.77, *p*-value: 0.02], and those with adverse reactions after the second and third vaccination dose [OR: 3.40, 95% CI: 2.37–4.88, *p*-value: <0.001], had a significantly higher possibility of reactive results, whereas those aged ≥65 years [OR: 0.28, 95% CI: 0.21–0.37, *p*-value: <0.001], having received heterologous booster vaccination (mRNA-1273 (Moderna)) [OR: 0.71, 95% CI: 0.55–0.91, *p*-value: 0.007], and suffering from hypertension [OR: 0.53, 95% CI: 0.40–0.71, *p*-value: <0.001], heart disease [OR: 0.63, 95% CI: 0.40–0.97, *p*-value: 0.0037], or diabetes mellitus [OR: 0.59, 95% CI: 0.38–0.94, *p*-value: 0.025] had a significantly lower possibility of reactive results (Table 2). In addition, using a multivariable analysis of all age groups, we identified that participants aged ≤40 years [aOR: 1.81, 95% CI: 1.19–2.75, *p*-value: 0.005], with adverse reactions after the second and third vaccination dose [aOR: 1.92, 95% CI: 1.19–3.09, *p*-value: 0.007], and aged ≥65 years and with adverse reactions after the second and third vaccination dose [aOR: 2.40, 95% CI: 1.24–4.61, *p*-value: 0.01] had a significantly higher possibility of reactive results.

### 3.5. T-SPOT.COVID Test and IgG(S) of Previously Infected Participants

We observed that five participants had mild sympathetic infection as documented during the interview at blood collection in March. In particular, we detected that the median (interquartile) of T-SPOT.COVID, neutralizing antibody, and IgG(S) antibody titers were 9 (3–43, 62–68), 500 (500–500) and 8657 (5372–9886), respectively. We found that two participants did not have reactive results in the T-SPOT.COVID test, although both had adverse reactions after the second or third dose and high levels of Nab antibody and IgG(S) antibody titers (Appendix A). Notably, a 21-year-old woman without any comorbidities had non-reactive result in the T-SPOT.COVID test.

## 4. Discussion

To the best of our knowledge, this was the first study to evaluate cellular immunity at the population level after booster vaccination against SARS-CoV-2. We found that 64.3% (700/1089) of participants had reactive results in the T-SPOT.COVID test, with cellular immunity being significantly reactive in those under 40 years of age and with adverse reactions after vaccination. Notably, despite having IgG(S) and neutralizing antibody titers of 500 AU/mL or higher, 33.9% (349/1031) and 33.5% (341/1017) of participants, respectively, did not have reactive results in the T-SPOT.COVID test. In addition, the distribution of T-SPOT.COVID in previously infected and uninfected individuals did not show a clear difference, suggesting that T-SPOT.COVID tests might not detect infection from variants of concern (VOCs). The main limitations of this study were as follows: Firstly, because only five (0.5%) of participants were previously infected, we could not fully compare cellular immunity between previously infected and uninfected subjects, or include cases of severe disease. Secondly, as all participants received a booster vaccination, we could not compare between the second and third doses of vaccination.

The distribution of T-SPOT.COVID in this study (Figure 1A) was almost identical to that of previous studies, showing that approximately 60% of participants had activated cellular immunity [35,36,37]. In detail, the proportion of individuals with activated cellular immunity in this study was approximately 10–20 points higher than that of individuals after two vaccination doses (47.5% (N = 95) [35], 54% (N = 13) [36], 46.2% (N = 91) [37]), as well as five points higher than that of those having received booster vaccination (59.6% (N = 47) [38]). Furthermore, ELISpot tests showed an increase in cellular responses after booster vaccination [35,38,39,40]. Hence, these findings suggested that booster vaccination induced cellular immunity.

Notably, despite IgG(S) and neutralizing antibody titers of 500 AU/mL or higher, 33.9% (349/1031) and 33.5% (341/1017) of participants, respectively, did not have reactive results in the T-SPOT.COVID test. This finding suggested that there is no clear correlation between humoral and cellular immunity [41], although the antibody titer of those with reactive cellular immunity (59%) was significantly higher than that of those with non-reactive cellular immunity. Cellular immunity was significantly reactive in those under 40 years of age and with adverse reactions after the second and third vaccination in multivariable analysis, whereas cellular immunity was not significantly reactive in those with heterologous boosting and comorbidities (hypertension, heart disease, or diabetes mellitus) in univariate analysis. A previous study showed that the induction of cellular immunity was harder with increasing age [42], which was consistent with our results. The effects of adverse reactions on cellular immunity were found to be significant in some studies, but not in other studies [43]; however, they were significant in our study. Although heterologous boosting has been reported to induce stronger cellular immune responses [44,45], our study did not detect it as the predictor. In the subanalysis stratified through the response of T-SPOT.COVID (Appendix A), a higher proportion of non-reactivity was observed in the heterologous boosting with mRNA-1273 compared to the homologous booster. This finding could be partly influenced by the higher proportion of older population > 65 years old in heterologous booster recipients than the homologous administration. The presence of unidentified biases, aside from age distribution, in this cohort may also prevent the recognition of the heterologous booster as an inducer of cellular immunity. Moreover, cellular immunity was not significantly reactive in comorbidities (hypertension, heart disease, or diabetes mellitus), thus increasing the severity of infection with SARS-CoV-2 and mortality [46,47,48,49,50]. These studies were consistent with our study, in that the days between booster vaccination and blood collection was not a significant factor in inducing cellular responses [51,52]. Other factors, such as blood type A, carrying a higher risk of SARS-CoV-2 infection and reinfection [53,54], smoking and drinking habits, or other high-risk factors in severe cases, were not significant factors in our study. It is known that lifestyle habits, such as smoking and alcohol consumption, can significantly lower the humoral immunity to COVID-19 and increase the mortality rate associated with COVID-19 [2,27]. The specific influence of lifestyle habits, including alcohol consumption, smoking, and drug use disorders, on the cellular immune response to COVID-19 remains unknown and warrants further investigation in future research.

Even though vaccination and history of previous infection induced stronger cellular immunity than vaccination alone [55,56], the spot numbers of previously infected and uninfected individuals did not differ significantly, suggesting several possibilities. Firstly, infection with VOCs might be less likely to affect the spot number because T-SPOT.COVID uses Wuhan antigens and, thus, has low detection sensitivity of VOC-induced cellular immunity. In particular, CD4- and CD8-positive T-cell responses to the Omicron variant were reported to make up 70–90% of those in Wuhan antigens [57,58,59,60]. Secondly, as all infected patients had mild infections, cellular immunity might not be an indicator of protection against breakthrough infection. Thirdly, comorbidities or previous infection with SARS-CoV1 could have influenced the results. For instance, two patients with small spot number had comorbidities, but their status regarding previous infection with SARS-CoV-1 was unknown. Notably, a 21-year-old woman without any comorbidities had no reactive results in the T-SPOT.COVID test. She had only weak symptoms at the time of infection but might have been immunodeficient.

Gao et al. reported that in a mouse model, repeated vaccination with a fifth or sixth dose caused a reduction in both humoral and cellular immune responses against Delta and Omicron variants. [61] However, there is still limited research on this issue in human subjects. We are continuing our investigation of humoral and cellular immune responses to the ancestral SARS-CoV-2 strain and its variants of concern in individuals who have received a fourth or subsequent dose within the same cohort. Our study will contribute to the development of effective vaccination strategies to optimize the number and type of vaccines.

It is unlikely that the radiation exposure after the Fukushima Daiichi Nuclear Powerplant Accident, which followed the Great East Japan Earthquake in 2011, influenced the study findings presented in this paper. The radiation exposure after the disaster was found to be low; the median annual external dose was estimated to be below 1 mSv from March to July 2011 [62], while the internal exposure dose for 99.986% of approximately 184,000 residents was less than 1 mSv [63]. In addition, a study performed among residents in the evacuated areas between June 2011 and March 2012 revealed that the distribution of white blood cell counts, including neutrophils and lymphocytes, showed no significant correlation with radiation dose exposure [64]. However, residents in evacuation areas, particularly those who have been displaced, were reported to have a higher risk of developing lifestyle-related diseases, including diabetes and cardiovascular diseases [65]. Therefore, we cannot deny the indirect effects of the disaster on the immune system as a contributing factor.

### Limitations

The study had certain limitations. Firstly, only five (0.5%) of participants were previously infected, which did not allow for a sufficient comparison between non-infected and infected individuals, or include any cases of severe infection. Secondly, there is a possibility that a certain number of asymptomatic or previously infected individuals were included in the study. Thirdly, antibody and T-SPOT.COVID tests were performed against the Wuhan antigens. As the effects of vaccine-induced humoral and cellular immunity are reduced against VOCs [66,67,68,69,70,71,72,73,74,75], antibody and T-SPOT.COVID tests for VOCs will be required in future studies. Fourthly, a false reactive result might be obtained for the T-SPOT.COVID test when tested in subjects previously exposed to SARS-CoV-1 or other similar coronavirus strains. Lastly, the cellular and humoral immune responses demonstrated in this study should be assessed in individuals without SARS-CoV-2 vaccinations and prior infections to ensure the absence of response.

## 5. Conclusions

This project is the first study to evaluate cellular immunity at the population level after booster vaccination against SARS-CoV-2. We found that 64.3% (700/1089) of participants had reactive results in the T-SPOT.COVID test, and that cellular immunity was significantly reactive in those under 40 years of age and with adverse reactions after vaccination. No statistically significant association was found between the cellular immune response and comorbidities such as hypertension, heart disease, or diabetes. Notably, despite IgG(S) and neutralizing antibody titers of 500 AU/mL or higher, 33.9% (349/1031) and 33.5% (341/1017) of participants, respectively, did not have reactive results in the T-SPOT.COVID test. The distribution of T-SPOT.COVID in previously infected and uninfected individuals did not show a clear difference, suggesting that T-SPOT.COVID testing might not detect infection from VOCs. In future studies, we need to investigate previously infected subjects and, in particular, their T-cell subsets.

## Figures and Tables

**Figure 1 vaccines-11-00920-f001:**
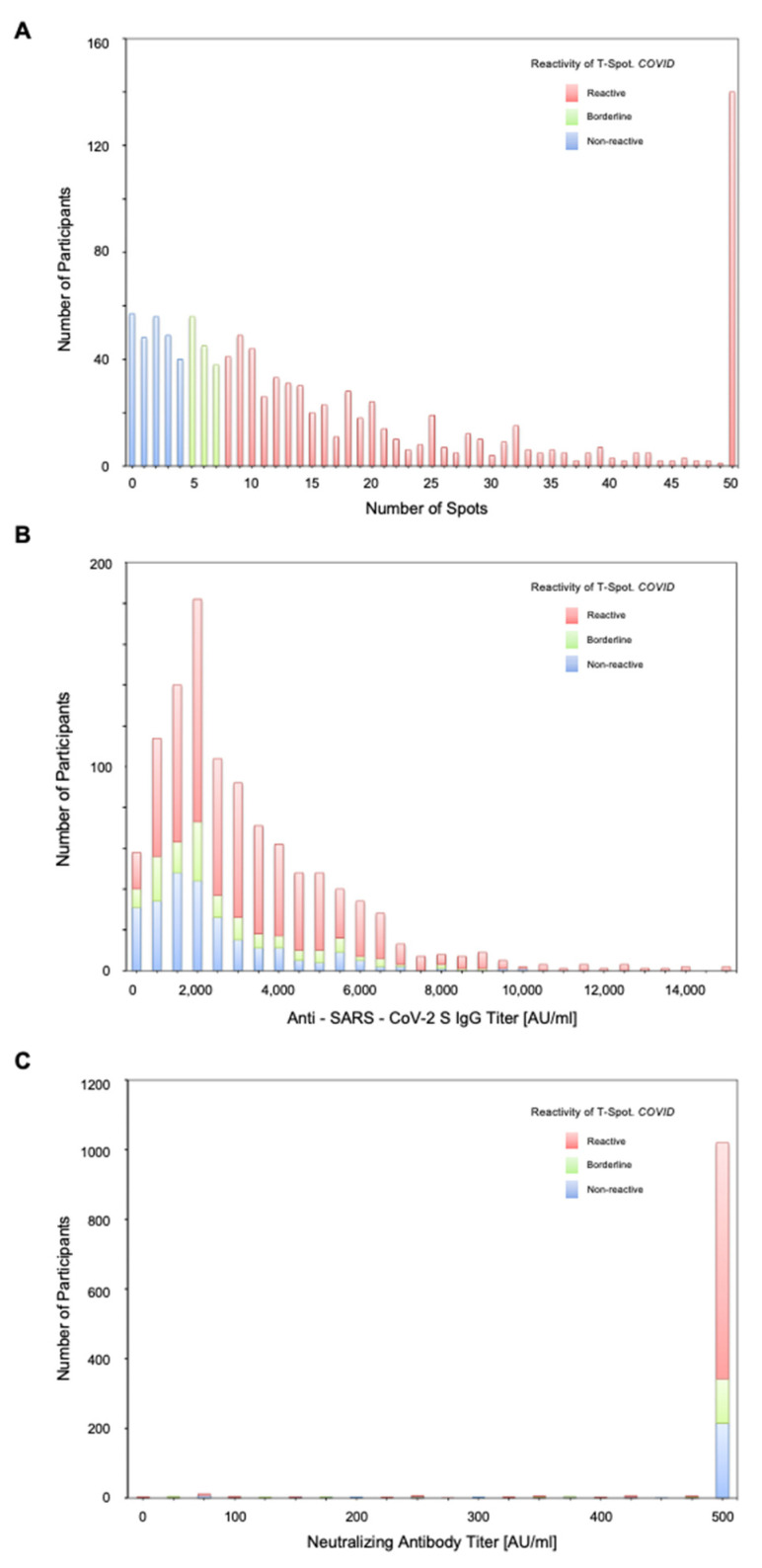
Histogram of T-SPOT.COVID, anti-SARS-CoV-2 IgG titer, and neutralizing antibody titer. (**A**) The number of spots in the T-SPOT.COVID test (**A**,**B**) IgG antibody titers against SARS-CoV-2 spike (S) protein (**B**,**C**) and neutralizing antibody titer (**C**). In (**A**), spots ≥ 50 is scored as 50. Spots ≤ 4, 5–7, and ≥8 were considered as non-reactive, borderline, and reactive, respectively. In (**C**), ≥500 AU/mL was scored as 500 AU/mL.

**Table 1 vaccines-11-00920-t001:** Descriptive data of interpretation of reactive T-SPOT.COVID test.

Variables	N = 1089
Age	56 (42–68)
<40 years	215 (19.7)
40–65 years	543 (49.7)
>65 years	331 (30.3)
Sex—Women	737 (67.5)
Blood type	
A	392 (35.9)
B	223 (20.4)
O	316 (28.9)
AB	95 (8.7)
Days between booster vaccination and blood collection	53 (29–65)
Less than 30 days	332 (30.4)
Between 30 and 59 days	367 (33.6)
More than 59 days	390 (35.7)
Vaccine type	
BNT162b2 (Pfizer–BioNTech)	682 (62.5)
mRNA-1273 (Moderna)	407 (37.3)
Previous infection	5 (0.5)
Anti-SARS-CoV-2 IgG titter (AU/mL)	2221 (1376–3966)
Smoking	194 (17.8)
Drinking Habits	406 (37.2)
Comorbidities—At least one	616 (56.4)
Medication—At least one	180 (16.5)
Adverse reactions after second and third vaccination dose—At least once	930 (85.2)
Adverse reactions after second vaccination dose—At least once	825 (75.5)
Adverse reactions after third vaccination dose—At least once	876 (80.2)

Median (interquartile) or number (percentage) are shown for continuous or categorical variables.

**Table 2 vaccines-11-00920-t002:** Results of logistic regression analysis of T-SPOT.COVID Test.

Variables	Reactive (n = 700)	Non-Reactive and Borderline (n = 389)	Univariable Analysis		Multivariable Analysis			
					All Ages Group		Over 65-Years-Old Group	
			OR (95% CI)	*p*-Value	aOR (95% CI)	*p* Value	aOR (95% CI)	*p* Value
Age								
<40 years	177 (25.3)	38 (9.8)	1.86 (1.25–2.77)	0.02	1.81 (1.19–2.75)	0.005	-	-
40–65 years	388 (55.4)	155 (39.8)	1 (ref)	<0.001	1 (ref)	<0.001	-	-
>65 years	135 (19.3)	196 (50.4)	0.28 (0.21–0.37)	<0.001	0.32 (0.22–0.46)	<0.001	-	-
Sex—Women	487 (69.6)	250 (64.3)	1.27 (0.98–1.65)	0.073	1.21 (0.87–1.69)	0.25	1.21 (0.64–2.3)	0.56
Blood type								
A	257 (38.2)	135 (38.2)	1 (ref)	0.60	1 (ref)	0.33	1 (ref)	0.33
B	139 (20.7)	84 (23.8)	0.87 (0.62–1.22)	0.42	0.84 (0.57–1.24)	0.39	0.60 (0.3–1.22)	0.16
O	211 (31.4)	105 (29.7)	1.06 (0.77–1.44)	0.74	0.98 (0.69–1.39)	0.90	0.61 (0.32–1.19)	0.15
AB	66 (9.8)	29 (8.2)	1.20 (0.74–1.94)	0.47	1.45 (0.84–2.50)	0.18	0.53 (0.22–1.27)	0.16
Days between booster vaccination and blood collection								
<30 d	212 (30.3)	120 (30.8)	1 (ref)	0.98	1 (ref)	0.75	1 (ref)	0.97
30–60 d	237 (33.9)	130 (33.4)	1.03 (0.76–1.41)	0.84	1.01 (0.69–1.50)	0.95	0.99 (0.45–2.18)	0.98
>60 d	251 (35.9)	139 (35.7)	1.02 (0.75–1.39)	0.89	0.88 (0.56–1.38)	0.58	0.91 (0.35–2.39)	0.85
Heterologous-booster (mRNA-1273)	241 (34.4)	166 (42.7)	0.71 (0.55–0.91)	0.007	1.19 (0.81–1.76)	0.37	1.34 (0.65–2.77)	0.42
Smoking	132 (19.2)	62 (16.4)	1.21 (0.97–1.68)	0.26	0.88 (0.60–1.30)	0.53	0.96 (0.38–2.43)	0.94
Drinking Habits	276 (40.6)	130 (35.1)	1.26 (0.97–1.64)	0.083	1.09 (0.80–1.50)	0.59	1.31 (0.67–2.55)	0.43
Comorbidities								
Hypertension	152 (21.8)	143 (36.9)	0.53 (0.40–0.71)	<0.001	1.06 (0.73–1.52)	0.77	1.03 (0.6–1.74)	0.93
Dyslipidaemia	59 (8.5)	36 (9.3)	1.12 (0.71–1.76)	0.63	1.11 (0.68–1.82)	0.68	0.67 (0.32–1.4)	0.29
Heart diseases	46 (6.6)	50 (12.9)	0.63 (0.40–0.97)	0.037	0.83 (0.49–1.43)	0.51	1.09 (0.52–2.26)	0.83
Diabetes mellitus	42 (6)	46 (11.9)	0.59 (0.38–0.94)	0.025	0.67 (0.39–1.15)	0.14	0.59 (0.26–1.33)	0.20
Adverse reaction after second and third dose	636 (91.9)	294 (77.0)	3.40 (2.37–4.88)	<0.001	1.92 (1.19–3.09)	0.007	2.40 (1.24–4.61)	0.010

Median (interquartile) or number (percentage) are shown for continuous or categorical variables. Definitions: 95% CI, 95% confidence interval; aOR, adjusted odds ratio; OR, odds ratio.

## Data Availability

The datasets analyzed during the present study are not publicly available.

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
