# Peer review of "Varying Cellular Immune Response against SARS-CoV-2 after the Booster Vaccination: A Cohort Study from Fukushima Vaccination Community Survey, Japan"

_vaccines, 2023, doi:10.3390/vaccines11050920_

Round 1

Reviewer 1 Report

The topic of the study is fascinating and well-conceived.

Some of the suggestions have been highlighted in a PDF file, and comments have been made using sticky notes.

The major limitations of the study have already been mentioned by the authors in a section.

I recommend a major revision.

Fine.

Author Response

Reviewer 1

  1. HCWs (Line 24); do not use abbreviation in abstract

Reply: We corrected the abbreviation to ‘healthcare workers.’ (Line 25)

  1. SARS-COV2 (Line 36)

Reply: We corrected it to ‘SARS-CoV-2.’ (Line 37)

  1. Eligibility criteria should be elaborated. Documented as being infected? Complex statement (Line 121)

Reply: We revised the descriptions in Section 2.2 of the Methods section to provide a clearer representation of participant recruitment and eligibility criteria, as detailed below;

‘2.2 Vaccination schedules, participant eligibility, and sample collection

Participants were recruited from Hirata village in Fukushima Prefecture in Japan as a part of the Fukushima cohort study [28–33]. In Hirata village, individuals were administered the first and second doses of the BNT162b2 vaccine (Pfizer/BioNTech, New York, USA) during two periods: April 6th to August 12th, 2021, for the initial dose, and April 27th to September 8th, 2021, for the subsequent dose. The third dose with either BNT162b2 (Pfizer-BioNTech) or mRNA1273 (Moderna, Cambridge, UK) was administered between December 23rd, 2021, and April 3rd, 2022.

            Participants eligible for this study were those who completed a booster vaccination and had a valid T-SPOT.COVID test result.

Blood samples were collected once during each designated period: June, September, and December of 2021, as well as March 2022. Blood collection (8 mL) was performed at medical facilities. The whole blood and serum samples were sent to the University of Tokyo (Tokyo, Japan) to measure SARS-CoV-2 specific antibodies and cellular immunity.

Information on age, sex, blood type, days between vaccination and blood collection, vaccine type, smoking, drinking habits, commodities, medication, and adverse reactions after vaccination was retrieved from the paper-based interview sheet (summarized in Table S1 and S4).’ (Lines 79–97)

Then, we modified the sentence indicated as follows;

‘A total of 1,089 individuals participated in this study. Five (0.5%) had a history of COVID-19 infection.’ (Line 127)

  1. Indicate ‘[21–21]’ (Line 127)

Reply: The square brackets showed the interquartile range. We changed the description with parenthesis since the brackets in this journal show the references. Of note, the interval between the first and second doses in most study participants was 21 days, resulting in a tiny interquartile range.

‘The median (interquartile range) interval between the first and second vaccination dose was 21 d (21-21), whereas that between the 127 second and third vaccination dose was 231 d (218-241).’ (Line 132–134)

  1. Author should discuss the comparative analysis when the booster is different. (Line 197)

Reply: Thank you for your insights on the heterologous booster administration in our cohort. We briefly compared the participant characteristics between the heterologous booster administration, and found a significant difference in age distribution.

‘Of note, a significant difference in age groups was observed between those receiving BNT162b2 and mRNA-1273 vaccines for booster administration. (p with Pearson's chi-square test <0.001); the numbers (percentages) in <40 years, 40–65 years, and >65 years old were 150 (22.0), 391 (57.3), and 141 (20.7) in BNT162b2, and 65 (16.0), 152 (37.3), 190 (46.7) in mRNA-1273 group, respectively.’ (in the table legends of Table S4)

Then, we discussed as follows;

‘Although heterologous boosting has been reported to induce stronger cellular immune responses [37,38], our study did not detect it as the predictor. In the subanalysis stratified by the response of T-SPOT.COVID (Table S4), a higher proportion of non-reactivity was observed in the heterologous boosting with mRNA-1273 compared to the homologous booster. This could be partly influenced by the higher proportion of older population >65 years old in heterologous booster recipients than the homologous administration. The presence of unidentified biases, aside from age distribution, in this cohort may also prevent the recognition of the heterologous booster as an inducer of cellular immunity.’ (Line 239–247)

  1. Study should be supplemented with controls. (Line 258)

Reply: We added another limitation as the reviewer suggested.

‘Lastly, the cellular and humoral immune responses demonstrated in this study should be assessed in individuals without SARS-CoV-2 vaccinations and prior infections to ensure the absence of response.’ (Line 305–307)

Of note, we showed the characteristics and results of SARS-CoV-2-specific antibodies and cellular immunity in Table S3, and added the descriptions to as follows;

‘We observed one participant with a nonreactive T-SPOT.COVID result, one with a borderline result, and three with reactive results. These findings align with the cohort without a history of COVID-19 infection (n =1,084), in which 249 (22.9%) individuals had nonreactive results, 138 (12.7%) had borderline results, and 697 (64.3%) had reactive results. In humoral immune response, all individulas with a history of COVID-19 infection showed strong reactivity of neutralizing antibody titer > 500 AU/ml and anti-SARS-CoV-2 IgG >5,000 AU/mL.’

Reviewer 2 Report

Cohort database assesed humoral and cellular immunity in 1089 participants.Well designed study."Cellular immunity was not reactive in those with comorbidities." is one of the striking findings of the study.

Author Response

Reviewer 2

Cohort database assessed humoral and cellular immunity in 1089 participants. Well-designed study. “Cellular immunity was not reactive in those with comorbidities.” is one of the striking findings of the study.

Reply: Thank you for your supportive comments and in consideration of your feedback, we have made modifications to the conclusion section to reflect the results of the multivariate analysis on comorbidities.

‘and that cellular immunity was significantly reactive in those under 40 years of age and with adverse reactions after vaccination. No statistically significant association was found between the cellular immune response and comorbidities such as hypertension, heart disease, or diabetes.’ (Lines 311–313)

Reviewer 3 Report

The study of Yuta Tani and their co-authors is well-written and valued.  There are a few minor comments which will help to improve the manuscript.

1. Despite the mention that, smoking and drinking habits were not significant factors in the study, it worthen discussed how alcohol and drug use disorder, as well as smoking, could affect the immune response, effectiveness, and safety of booster doses of vaccines. 

2. The study by Gao (iScience, 2022) reported that the repeated use of vaccine boosters induced humoral and cellular tolerance against the Delta and Omicron variants. It should be mentioned in the discussion.

3. It may be interesting for readers to devote a couple of sentences to the radiological situation of the area and how the accident at the nuclear power plant in 2011 affected the immunity of local residents, and how this may affect vaccination.

Author Response

Reviewer 3

The study of Yuta Tani and their co-authors is well-written and valued. There are a few minor comments which will help to improve the manuscript.

  1. Despite the mention that, smoking and drinking habits were not significant factors in the study, it worthen discussed how alcohol and drug use disorder, as well as smoking, could affect the immune response, effectiveness, and safety of booster doses of vaccines.

Reply: Thank you for your suggestion. We added the discussion as follows;

‘Other factors, such as blood type A, carrying a higher risk of SARS-CoV-2 infection and reinfection [46,47], smoking and drinking habits, or other high-risk factors in severe cases, were not significant factors in our study. It is known that lifestyle habits such as smoking and alcohol consumption can significantly lower the humoral immunity to COVID-19 and increase the mortality rate associated with COVID-19 [2,27]. The specific influence of lifestyle habits, including alcohol consumption, smoking, and drug use disorders, on the cellular immune response to COVID-19 remains unknown and warrants further investigation in future research.’ (Line 252-259)

  1. The study by Gao (iScience, 2022) reported that the repeated use of vaccine boosters induced humoral and cellular tolerance against the Delta and Omicron variants. It should be mentioned in the discussion.:

Reply: Thank you for your insights. We added the discussion as follows;

‘Gao et al. reported that in a mouse model, repeated vaccination with a 5th or 6th dose caused a reduction in both humoral and cellular immune responses against Delta and Omicron variants. [69] However, there is still limited research on this issue in human subjects. We are continuing our investigation of humoral and cellular immune responses to the ancestral SARS-CoV-2 strain and its variants of concern in individuals who have received a 4th or subsequent dose within the same cohort discussed here. Our study will contribute to the development of effective vaccination strategies to optimize the number and type of vaccines.’ (Lines 274–281)

  1. It may be interesting for readers to devote a couple of sentences to the radiological situation of the area and how the accident at the nuclear power plant in 2011 affected the immunity of local residents, and how this may affect vaccination.

Reply: We added the discussion as follows;

‘It is unlikely that the radiation exposure after Fukushima Daiichi Nuclear Powerplant accident following the Great East Japan Earthquake in 2011 influenced the study finding presented here. The radiation exposure after the disaster was found to be low; the median annual external dose was estimated to be below 1 mSv from March to July 2011 [70], and the internal exposure dose for almost all the residents, 99.986% of approximately 184,000 residents, was less than 1 mSv [71]. In addition, a study performed among residents in the evacuated areas between June 2011 and March 2012 revealed that the distribution of white blood cell counts, including neutrophils and lymphocytes, showed no significant correlation with radiation dose exposure [72]. However, residents in evacuation areas, particularly those who have been displaced, have been reported to have a higher risk of developing lifestyle-related diseases, including diabetes and cardiovascular diseases [73]. Therefore, we cannot deny the indirect effects of the disaster on the immune system as a contributing factor.’ (Lines 282–294)

Round 2

Reviewer 1 Report

Authors have revised the MS